# Inflammatory Bowel Disease Guidelines for Corneal Refractive Surgery Evaluation

**DOI:** 10.3390/jcm11164861

**Published:** 2022-08-19

**Authors:** Majid Moshirfar, David A. Fuhriman, Amir Ali, Varshini Odayar, Yasmyne C. Ronquillo, Phillip C. Hoopes

**Affiliations:** 1Hoopes Vision Research Center, Hoopes Vision, 11820 S. State St., Ste. 200, Draper, UT 84020, USA; 2John A. Moran Eye Center, University of Utah School of Medicine, Salt Lake City, UT 84132-2101, USA; 3Utah Lions Eye Bank, Murray, UT 84107, USA; 4McGovern Medical School, The University of Texas Health Science Center at Houston, Houston, TX 77030, USA; 5John Sealy School of Medicine, The University of Texas Medical Branch at Galveston, Galveston, TX 77555, USA; 6Department of Molecular and Cellular Biology, Harvard University, Cambridge, MA 02138, USA

**Keywords:** inflammatory bowel disease, IBD, ulcerative colitis, Crohn’s disease, corneal refractive surgery, LASIK, PRK, SMILE, implantable collamer lens

## Abstract

Inflammatory bowel disease (IBD) is a chronic systemic inflammatory condition that can potentially adversely affect surgical outcomes in patients receiving elective ophthalmic procedures. In this case series, 21 eyes of 11 patients with ulcerative colitis or Crohn’s disease underwent laser in situ keratomileusis (LASIK), photorefractive keratectomy (PRK), or small incision lenticule extraction (SMILE). Their surgical outcomes were followed up for an average of 8.9 ± 4.6 months. All the patients in this study did well, with 100% of eyes corrected for distance vision achieving uncorrected distance visual acuity 20/20 by postoperative month three. Common symptoms noted during the postoperative period included dry eyes, irritation, foreign body sensation, and blurry vision, all of which improved in prevalence and severity over the follow-up period, and none of the patients experienced a flare-up of their disease. Despite the successful outcomes in these patients, the authors recognize the inherent risks of operating on patients with IBD. Currently, there are no consensus guidelines for clinicians to follow to ensure that they are adequately screening these patients for eligibility, so the authors are suggesting a relevant, focused review of systems, a brief IBD history-related questionnaire, and a preliminary surgical decision-making flowchart for use in surgical evaluation.

## 1. Introduction

Inflammatory bowel disease (IBD) is a category of systemic inflammatory autoimmune disease processes affecting the gastrointestinal (GI) tract, divided into two major categories: Crohn’s disease (CD) and ulcerative colitis (UC). IBD has a prevalence of 1300 per 100,000 persons in the US and 84.3 per 100,000 globally [1,2]. There are various GI histopathological differences between UC and CD; however, there are similarities in their extraintestinal manifestations (EIMs), including the ocular manifestations which affect 2–7% of IBD patients [3].

Due to the complex systemic nature of IBD, the ocular evaluation for elective ophthalmic procedures must be comprehensive to determine surgical eligibility. Clinicians need to be aware of common ocular manifestations of CD and UC such as anterior uveitis, scleromalacia perforans, and optic neuritis when caring for these patients. Chronic inflammation, as well as fat-soluble vitamin deficiencies (especially hypovitaminosis A) in chronic IBD also commonly lead to dry eye symptoms. Dry eye is a significant factor to consider in patients desiring corneal refractive surgery, as these procedures carry an inherent risk of increased chronic dry eye postoperatively. Additionally, ophthalmologists must consider the immunomodulating effects of the medications commonly used in the treatment of IBD, including azathioprine, tacrolimus, and methotrexate. Most importantly, clinicians need to be able to recognize the stability of the patient’s disease, as an uncontrolled disease state may influence and exacerbate the immune response to various triggers such as ocular surgery, resulting in unfortunate outcomes such as necrotizing keratitis [4].

This study presents a five-year case series in an ophthalmologist’s clinic of patients with IBD that have undergone laser in situ keratomileusis (LASIK), photorefractive keratectomy (PRK), and small incision lenticule extraction (SMILE) refractive surgery. We look into the patient demographics, medical history, stability of disease, preoperative findings, and postoperative findings and then discuss the most common ophthalmic EIMs and steps that ophthalmologists can take to ensure patient surgical candidacy. To the best of our knowledge, this is the first case series of ocular refractive surgery in the IBD patient population to be reported in the English language in the ophthalmic literature.

## 2. Methods

This study was conducted in accordance with the tenets of the Helsinki Declaration of 1964, as revised in 2013. It was also Health Insurance Portability and Accountability Act of 1996 (HIPAA)-compliant. All subjects gave their informed consent for inclusion before their records were examined for this study. This study was approved by the Hoopes Vision Ethics Board and BRANY IRB #20-12-547-823 (New York) per research standards and state law.

This study is a retrospective case series of 11 patients with diagnoses of UC or CD who received PRK, LASIK, or SMILE performed by a single surgeon at a single tertiary eye center between the years 2018 and 2022. The patients were identified by a search into the patient records for correspondence with a gastroenterologist regarding ulcerative colitis or Crohn’s disease before undergoing corneal refractive surgery. This search yielded 19 patients (8 with UC, 11 with CD), of whom 11 received LASIK, PRK, or SMILE and were included in our study. See Figure 1 for inclusion and exclusion of these patients. LASIK flap creation was performed with either a Zeiss VisuMax Femtosecond Laser or an Alcon WaveLight FS200 Femtosecond Laser. LASIK and PRK stromal tissue ablation were performed with an Alcon WaveLight EX500 Excimer Laser. SMILE lenticule creation was performed with a Zeiss VisuMax Femtosecond Laser.

This case series analyzes those 11 patients: (a) who were eligible for corneal refractive surgery, (b) whose records were obtainable, and (c) who actually underwent corneal refractive surgery (Figure 1). Data compiled from the medical records are included in Table 1 and Appendix A. The primary outcomes of the chart reviews included preoperative values for uncorrected distance visual acuity (UDVA); best-corrected distance visual acuity (BDVA); manifest refraction (MRx); and all postoperative values for UDVA, BDVA, and MRx. Secondary outcomes included patient-reported symptoms.

## 3. Results

### 3.1. Patient Demographics and Patient Selection

From the 19 patients with IBD evaluated for refractive surgery over five years, Table 1 lists the demographic information for 14 patients, excluding the patients with unavailable records (2), those not eligible for surgery (2), and the patient receiving an ICL (1). Of the 14 patients (average age of 36.6 years), 6 were female and 6 were diagnosed with UC. The average number of years since diagnosis and since the last gastrointestinal flare, if available, was 10.5 years and 6 years, respectively. Nine patients were on monotherapy, two were on dual therapy, and two were on triple therapy. The most common immunotherapies were azathioprine (purine synthesis inhibitor) and mesalamine (5-aminosalicylic acid (ASA)) with four patients each, followed by infliximab (TNF-alpha inhibitor), adalimumab (TNF-alpha inhibitor), and vedolizumab (integrin receptor antagonist) with three patients on each of these therapies. Sulfasalazine (5-ASA prodrug) was taken by only two patients.

One patient with UC underwent colectomy and three patients with CD had undergone some degree of bowel resection. The most common non-IBD medical conditions were allergies and migraines present in six and four patients, respectively. Migraines may be a side effect of immunotherapy. One 28-year-old male patient had a history of arthritis: a potential EIM of IBD.

### 3.2. Ocular History and Preoperative Evaluation

To assess prior ocular manifestations of IBD, the patient’s ocular history was documented and assessed, as shown in Table 1. Patients were previously diagnosed with keratoconjunctivitis sicca (1), episcleritis (1), iritis (1), conjunctivitis (3), contact-lens induced keratopathy (1), MGD (2), chalasis (1), posterior embryotoxon (1), papilloma (1), and congenital retinal fold resulting in amblyopia (1). Due to ocular examination findings, two patients were excluded from laser refractive surgery. Appendix A shows preoperative testing results including K1, K2, pachymetry, ablation depth, residual stromal thickness, and preoperative UDVA, manifest refraction, and BDVA for each patient that received corneal refractive surgery. Preoperatively for the 21 eyes, the average K1, K2, and pachymetry were 44.2 ± 1.1 D, 45.1 ± 0.95 D, and 547.6 ± 20.23 μm, respectively. For patients with PRK and LASIK, the average postoperative ablation depth and residual stromal thickness were 52.5 ± 16.81 μm and 395 ± 27 μm. The average preoperative UDVA was greater than 20/300 ± 200 (range: 20/125–20/1000), and preoperative BDVA was 20/20 ± 0.0.

### 3.3. Primary Outcomes

The visual outcomes after laser refractive surgery were successful as evidenced by the postoperative examinations over the follow-up period, listed in Figure 2 and Appendix A. Preoperatively, 0% of eyes had a UDVA better than 20/40 and 100% of eyes had BDVA 20/40 or better. On postoperative day 1, 95% of eyes had a UDVA better than 20/40 and by postoperative month 1, 100% of eyes had achieved UDVA 20/40 or better, with 94% at 20/20 UDVA and 100% at 20/20 BDVA or better. At the postoperative month 3 follow-up, 100% of eyes had attained UDVA 20/20 or better. At the one-year follow-up, from the initial 21 eyes in the study, only 14 eyes had been followed up. Of those patients, 100% of eyes had UDVA 20/40 or better, and 86% of eyes had achieved 20/20. Of note, 2 eyes had regressed from UDVA 20/20 and 20/15 to 20/25 by postoperative month 12, which was presumed to be due to dry eye. The BDVA remained 20/20 or better for 100% of our patients at the one-year follow-up (see Figure 2).

### 3.4. Secondary Outcomes

In addition to the primary visual acuity outcomes, we were also interested in looking into patient symptoms postoperatively as detailed in Figure 3. The most common patient symptoms on postoperative day 1 were dry eye in 5 patients, foreign body sensation (FBS) in 3 patients, and irritation in 2 patients. The treatment of choice for dry eyes and FBS was to increase the frequency of preservative-free artificial tear (PFAT) use. The frequency of PFAT application was increased to as high as every 30 min for patients with moderate levels of dryness or mild FBS. At one week follow-up, both the number of patients with dryness and the severity of the dryness were reduced with a resultant reduction in PFAT use. Two patients expressed difficulty with near VA and mild blurriness; however, that was presumed to be secondary to presbyopia and dryness. At 1 month follow-up, patient 8 expressed interest in corrective lenses to improve her mid-distance visual acuity and was prescribed mid-distance glasses. Patient 10 experienced fluctuating near vision, for which PFATs were prescribed every 2–3 h. At one year follow-up, these two patients reported resolution of all symptoms. At one-year follow-up, only one patient noted persistent morning dry eye for which PFATs were recommended as needed. Patient 9 was noted to have self-limiting viral adenoconjunctivitis (symptoms and exam findings were not consistent with HSV or VZV keratitis) around 10 months postoperatively, and examination at one-year follow-up showed lid and lash collarettes for which lid hygiene and warm compresses were recommended. These symptoms were likely not related to surgery. The most common patient symptom was dry eyes, which had improved significantly by postoperative month 3. Overall, the quantity and severity of patient symptoms decreased over the follow-up period.

## 4. Discussion

The patients we evaluated in our study have an autoimmune, systemic, and inflammatory disease; however, their corneal refractive surgery outcomes were successful. The average preoperative UDVA in our 21 eyes was worse than 20/300 ± 200 (range: 20/125–20/1000) and BDVA was 20/20. By postoperative month 3, all eyes had achieved UDVA 20/20 or better. The most common postoperative symptom was dry eyes, which occurred in 5 patients on postoperative day 1 and decreased to 1 patient by postoperative month 3. Although the patients in our study achieved successful outcomes, patients with IBD have a complex and multisystemic disease process that can adversely affect surgical outcomes. These patients were evaluated before their surgery and expressed disease stability, and prior to the operation, correspondence was made with their gastroenterologists to ensure disease quiescence on their current medication regimen. In addition, a thorough discussion of risks, benefits, and alternatives was discussed before proceeding with surgery.

There is currently no consensus guideline for ophthalmologists to consult when considering surgical treatment of patients with IBD. Therefore, as ophthalmologists, it is important to understand the basics of IBD pathophysiology and its common and serious EIMs (especially ocular manifestations). Additionally, it is important to be familiar with the medical management of IBD, side effects of treatment, and manifestations of GI and ocular flares to properly care for these patients. An understanding of these concepts is crucial in providing quality care and evaluating surgical risk for these patients. Even attempting surgery in patients with inactive disease can be dangerous, as the body’s heightened immune response can result in excessive inflammation. In rare cases, ocular surgery has led to disastrous complications including necrotizing keratitis, corneal melt, and hypopyon uveitis [4,5,6,7]. IBD has also been demonstrated to have a strong association with keratoconus [8,9]. Corneal topography is one of the fundamental components of the surgical evaluation for all patients desiring to undergo corneal refractive surgery, and a careful analysis of the imaging should detect any underlying keratoconus. More research needs to be performed to determine whether there is an elevated risk for postoperative ectasia in patients with IBD. Several of the immunomodulatory agents that IBD patients take to manage their condition can also confer surgical risks. For example, with immunosuppression, the risk of post-surgical infections or flare-ups of latent viral infections (such as herpes simplex virus or Varicella–Zoster virus) is elevated. It is advantageous and necessary on the part of ophthalmology and gastroenterology to establish a framework of guidelines and questions that would aid in the consideration of ophthalmological surgery such as corneal refractive surgery to minimize the risk of deleterious adverse events [10,11,12,13].

Recent studies explore how to pharmacologically speed up the corneal surgical wound healing process. This results in more rapid corneal re-epithelialization and decreased inflammation after corneal refractive surgery and may be beneficial in the context of chronic inflammatory states such as IBD [14,15]. When evaluating patients with IBD for candidacy in obtaining corneal refractive surgery, it is important to determine whether the disease is stable and in remission. In this paper, we propose a set of relevant review of system questions (ROS) for this patient population (Table 2) which may alert the clinician to a potential acute CD or UC general flare as well as many common EIM symptoms.

Acute gastrointestinal flares of UC and CD can manifest in many ways, as highlighted in Table 2. These can manifest as bloody or non-diarrhea, abdominal pain, weight loss, weakness secondary to anemia, bloating, fever, stricture formation in the intestines leading to bowel obstruction or stool variability, and intestinal fistula formation (e.g., enterovesicular, enteroenteral, enterocutaneous, enterovaginal). Gastrointestinal flares of IBD are commonly associated temporally with EIMs such as peripheral arthritis, aphthous ulcers, episcleritis, and erythema nodosum. On the other hand, other EIMs such as anterior uveitis, ankylosing spondylitis, and primary sclerosing cholangitis can flare up independent of active intestinal disease [3,16]. If the patient describes any current or recent symptoms that are suspicious or suggestive of an IBD flare, the patient should be referred to their gastroenterologist for an evaluation before pursuing surgery. When evaluating patients with IBD for elective surgeries, it is important to consider both intestinal and extraintestinal disease activity, as they may not be synchronous.

In addition to the ROS, we have developed a basic framework of focused questions regarding the patient’s IBD history (see Table 3). These questions should include pertinent details regarding the timing of the patient’s disease, current management of the disease (including the number and dosage of immunomodulatory therapies), any bowel surgeries and resultant nutritional deficiencies, and any complications from the disease including GI fistulae or strictures, which can indicate disease stability. Patient smoking habits should also be considered when evaluating IBD patients because heavy smoking has been demonstrated to increase the risk of CD flares, while smoking cessation has been shown to increase the risk of UC flares. Thus, ophthalmologists need to take note of any changes in routine while considering surgery for IBD patients.

Once a patient’s medical history, including IBD-related ROS and disease course, has been recorded, the ophthalmologist can decide whether or not to proceed with elective ocular procedures. In Figure 4, we detail a basic flowchart to guide ophthalmologists in determining when it is appropriate to move forward with surgery. The algorithm factors in important predictors such as disease course, ROS, time since diagnosis, acute flares, endoscopic procedures, and results of recent GI visits. Once the clinician and the patient determine to move forward with surgery, correspondence with the patient’s gastroenterologist is recommended to notify them of the decision and to allow them to voice any concerns.

We have retrospectively developed these screening tools with the intent to develop a more standardized approach in evaluating IBD patients for ocular surgical candidacy. These tools are beneficial in aiding the clinician in recognizing symptoms of active disease that would convey additional surgical risk in these patients. We recognize that these tools have not been validated, and they should be used only as a guide until they have been validated in future studies or until more official guidelines have been established. The authors hope to promote a more active interdisciplinary dialogue for guidelines regarding the care of patients with IBD as well as other autoimmune or inflammatory conditions. The screening tools proposed herein are to serve as a nidus for these conversations. It behooves the American Academy of Ophthalmology, perhaps in collaboration with the American Gastroenterological Association, to establish an official framework for decision-making regarding the ocular surgical management of IBD patients.

## Figures and Tables

**Figure 1 jcm-11-04861-f001:**
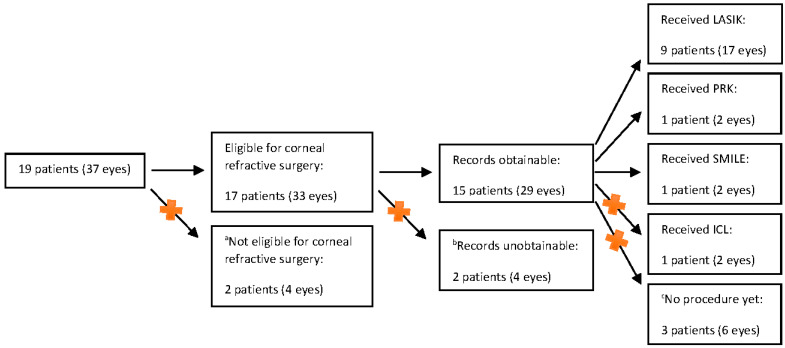
Patient selection flowchart. A red X indicates patients who were excluded from our data analysis. ^a^ 1 patient was ineligible due to contact lens-induced keratopathy; 1 patient was ineligible due to anterior basement membrane changes; ^b^ 2 patients were followed postoperatively by an outside provider, and they opted not to participate in this study; ^c^ 3 patients were eligible for corneal refractive surgery but have not yet scheduled the procedure due to patient preference.

**Figure 2 jcm-11-04861-f002:**
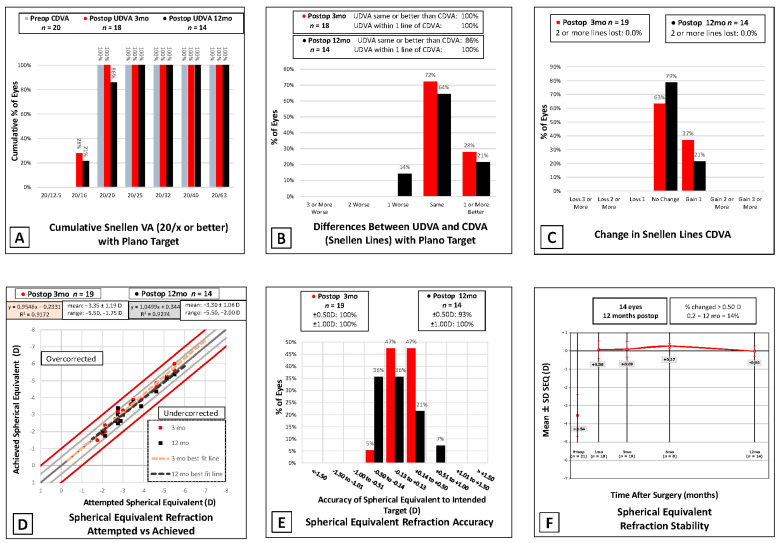
Six standard graphs for reporting corneal refractive surgical outcomes. (**A**) Postoperative uncorrected distance visual acuity versus preoperative corrected distance visual acuity. (**B**) Differences between uncorrected distance visual acuity and corrected distance visual acuity postoperatively. (**C**) Change in corrected distance visual acuity preoperatively versus postoperatively. (**D**) Spherical equivalent refraction attempted versus achieved. (**E**) Postoperative spherical equivalent refraction accuracy. (**F**) Stability of spherical equivalent refraction over time. Preop: preoperative; postop: postoperative; SEQ: spherical equivalent refraction.

**Figure 3 jcm-11-04861-f003:**
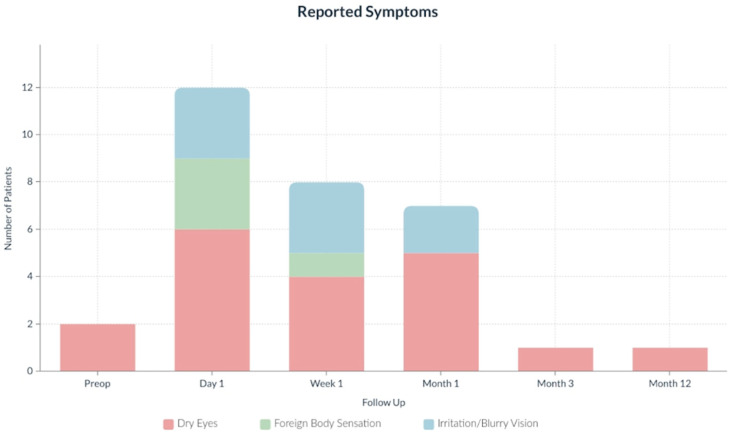
Patient-reported symptoms. Symptoms decreased in frequency and severity over the follow-up period.

**Figure 4 jcm-11-04861-f004:**
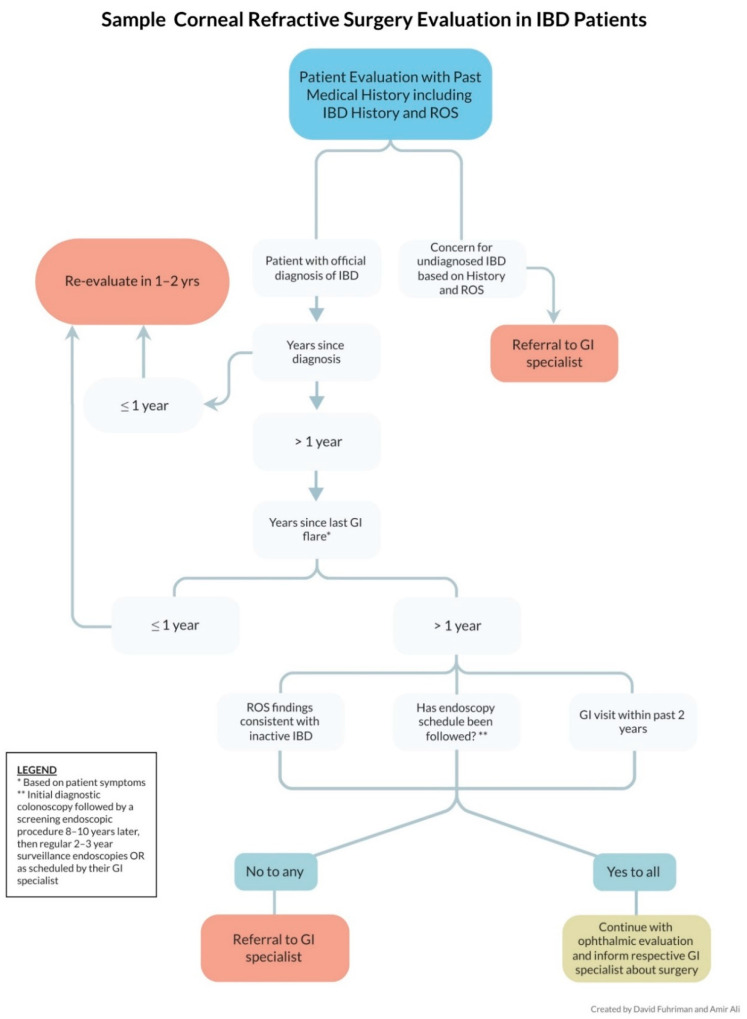
Sample corneal refractive surgery evaluation in IBD patients.

**Table 1 jcm-11-04861-t001:** Patient demographics.

Patient	Age	Sex	IBD Diagnosis	Years Since Diagnosis	Years Since Last Flare	Immuno-therapy	Associated Procedures	Medical History	Ocular History
1	35	M	UC	13	unk	Azathioprine	Colectomy	Migraines, Allergies	Myopia, Allergic Conjunctivitis
2	38	F	UC	unk	6	Infliximab	none	Hypertension Allergies	CMA
3	41	F	UC	unk	unk	Adalimumab	none	Migraines, Depression	CMA, Congenital Retinal Fold OS, Amblyopia OS
4	40	F	UC	unk	unk	Azathioprine, Sulfasalazine	none	Migraines, Depression, Hyperlipid-emia	CMA
5	41	M	UC	1	unk	−	none	Allergies	CMA, KCS
6	39	M	UC	unk	5	Sulfasalazine	none	Proctitis	CMA
7	30	M	CD	10	unk	Infliximab	Bowel Resection	none	Myopia, Giant Papillary Conjunctivitis, Episcleritis
8	46	F	CD	27	unk	Vedolizumab	none	Allergies	CMA, Presbyopia, Iritis, Viral Conjunctivitis, Papilloma OD
9	28	M	CD	14	unk	Vedolizumab, Mesalamine, Azathioprine	none	Arthritis	Myopia, Chalazion s/p Excision, MGD, Bilateral Ptosis
10	35	M	CD	6	unk	Mesalamine	none	none	Myopia, Posterior Embryotoxon
11	37	F	CD	8	6	Azathioprine, Mesalamine, Adalimumab	none	none	Myopia
12 ^a^	32	M	CD	14	7	Infliximab	Ileocecal Resection	none	CMA
13 ^a^	33	F	CD	2	unk	Vedolizumab	none	Allergies	CMA, Contact Lens-Induced Keratopathy, Mild MGD OD, Minimal Chalasis
14 ^a^	38	M	CD	unk	unk	Adalimumab, Mesalamine	Bowel Resection	Migraines, Allergies	CMA

UC: ulcerative colitis; unk: unknown; CD: Crohn’s disease; CMA: compound myopic astigmatism; OS: left eye; KCS: keratoconjunctivitis sicca; OD: right eye; MGD: meibomian glandular dysfunction. ^a^ Patients who are qualified for corneal refractive surgery but have not undergone the procedure.

**Table 2 jcm-11-04861-t002:** Focused ROS questions for IBD patient evaluation. This table details relevant questions regarding symptoms that should be addressed when completing a review of systems for patients with IBD. ROS: review of systems; GI: gastrointestinal.

ROS Questions for IBD Patient Evaluation	
**General**	Have you had any recent fevers?
	Have you noticed any recent weight loss?
	Are you experiencing body aches?
	Have you noticed abnormal muscle weakness?
	Do you have oral (aphthous) ulcers?
	Do you have joint pain (hip, buttocks, thigh, knee)?
	Have you been recently diagnosed with arthritis?
	Do you have recent lower back pain (lower, middle, upper)?
	Do you have tender nodules on your shins?
**GI**	Do you have cramping or pain in your abdomen?
	Do you have abnormal bowel movements such as loose, hard, or minimal stool (ex: pencil-thin stool)?
	Do you have diarrhea (bloody or non-bloody)?
	Do you have hematochezia (red bloody stool)?
	Do you have melena (black stool)?
	Do you have difficulty controlling bowel movements?
	Do you have rectal pain or irritation when sitting down, with movements, or when coughing?
	Do you have skin irritation around the anogenital region?
	Do you have redness, warmth, or swelling around the anus?
**Ocular**	Do you have blurred vision?
	Have you had a recent loss of vision?
	Do you have excessive tearing?
	Do you have eye redness?
	Do you have eye pain?
	Do you have dry eyes?
	Do you have itchy eyes?
	Do you have light sensitivity?

**Table 3 jcm-11-04861-t003:** IBD history questionnaire. This figure details relevant questions regarding the disease course for a patient with IBD. VZV: Varicella–Zoster virus; HSV: herpes simplex virus.

IBD History	
**Disease timing**	What was your age at diagnosis? How many years have you had UC or CD?
	When was your last GI flare?
**Medical management**	When was your last gastroenterology appointment?
	When was your last colonoscopy? What were the results of your last colonoscopy?
	Which medications are you currently taking? When was your last medication change? Why was it changed?
	Do you have any vitamin malabsorption? Are you taking any vitamins or supplements?
**Surgical management**	Have you undergone any GI procedures? If so, do you have an ostomy?
**Disease complications**	Do you currently have an intestinal or perianal fistula? Have you ever had an intestinal or perianal fistula?
	Do you currently have an intestinal stricture? Have you ever had an intestinal stricture?
**Family history**	Do you have a family history of IBD? If so, what is the disease status of those family members?
**Operative concerns**	Do you have a history of VZV or HSV keratitis?
	Have you ever had a diagnosis of uveitis, scleritis, episcleritis, or conjunctivitis? Do you have dry eyes? Do you have a history of keratoconus?
	Do you have a history of tobacco use? If so, how many packs per day? Are you currently trying to quit tobacco use?

## Data Availability

Data used in the analysis of this paper can be found in Appendix A.

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
