# Peer review of "Inflammatory Bowel Disease Guidelines for Corneal Refractive Surgery Evaluation"

_jcm, 2022, doi:10.3390/jcm11164861_

Round 1
Reviewer 1 Report
The study is well conducted, the subject is interesting and underlines the importance of the preoperative assessment in refractive surgery
Nevertheless, a link between keratoconus/ keratoconus suspect and IBD has ben yet proven. The analysis of the corneal topography being at the center of the preoperative assessment, it needs to be highlighted and discussed in your article.
Reviewer 2 Report
I read with great interest this manuscript title: Inflammatory bowel disease guidelines for corneal refractive surgery evaluation where the authors elegantly describe a series of cases in which some guidelines are recommended for the management of patients with Inflammatory bowel who want refractive surgery. I totally agree with reporting this kind of case series. A review has recently been published on the update of indications and contraindications in corneal refractive surgery (DOI: 10.1016/j.oftal.2022.07.001) on topics similar to the one presented by the authors. Therefore, I congratulate authors for this work
Just Minor changes:
Authors should use the acronyms after the first time they mention a phrase. In this case in section 3.3 there is an error:
The treatment of choice for dye eyes and FBS was to increase the frequency of PFAT use. The frequency of preservative-free artificial tear (PFAT) application was increased to as high as every 30 minutes for patients with moderate levels of dryness or mild FBS.
I strongly agree with the questionnaires developed by the authors, as the authors point out, these must be validated. I suggest highlighting as a limitation that although they have served as a guide, the tests (questionnaires) must be validated in a separate study.
Reviewer 3 Report
the manuscript is very well written and deals with a very current topic and I recommend some changes. I recommend extending the introductory part by specifying what are the consequences on the ocular surface of systemic chronic inflammatory diseases. even if it is intuitive for the reader it would be better to specify them before passing the discussion on the post-operative. I recommend simplifying the chapter on materials and methods to make speech more fluid. I also recommend to insert, if the authors agree, some bibliographic notes both in the text (after note 13 on line 207). Meduri A, Grenga PL, Scorolli L, Ceruti P, Ferreri G. Role of cysteine in corneal wound healing after photorefractive keratectomy. Ophthalmic Res. 2009;41(2):76-82. doi: 10.1159/000187623. Epub 2008 Dec 20. PMID: 19122468. Meduri A, Scorolli L, Scalinci SZ, Grenga PL, Lupo S, Rechichi M, Meduri E. Effect of the combination of basic fibroblast growth factor and cysteine on corneal epithelial healing after photorefractive keratectomy in patients affected by myopia. Indian J Ophthalmol. 2014 Apr;62(4):424-8. doi: 10.4103/0301-4738.119420. PMID: 24145571; PMCID: PMC4064216.
Meduri A, Scalinci SZ, Morara M, Ceruti P, Grenga PL, Zigiotti GL, Scorolli L. Effect of basic fibroblast growth factor in transgenic mice: corneal epithelial healing process after excimer laser photoablation. Ophthalmologica. 2009;223(2):139-44. doi: 10.1159/000187686. Epub 2008 Dec 18. PMID: 19092284.
Scorolli L, Meduri A, Morara M, Scalinci SZ, Greco P, Meduri RA, Colombati S. Effect of cysteine in transgenic mice on healing of corneal epithelium after excimer laser photoablation. Ophthalmologica. 2008;222(6):380-5. doi: 10.1159/000151691. Epub 2008 Aug 28. PMID: 18753800. these are very interesting studies on how to pharmacologically speed up corneal re-epithelialization after excimer laser. this greatly improves the inflammatory cascade and the risk of haze. Very useful if the patient has a chronic inflammatory disease. I recommend English language improvements. I recommend more emphasis on conclusions
